# Diversity, Succession and Seasonal Variation of Phylloplane Mycoflora of *Leucaena leucocephala* in Relation to Its Leaf Litter Decomposition

**DOI:** 10.3390/jof8060608

**Published:** 2022-06-06

**Authors:** Saloni Gulati, P. Chitralekha, Manisha Arora Pandit, Roma Katyal, Neeru Bhandari, Poonam Mehta, Charu Dogra Rawat, Surinder Kaur, Jasleen Kaur

**Affiliations:** 1Department of Botany, Dyal Singh College, University of Delhi, Delhi 110003, India; salonigulati@dsc.du.ac.in (S.G.); pchitralekha@dsc.du.ac.in (P.C.); romakatyal@dsc.du.ac.in (R.K.); neerubhandari@dsc.du.ac.in (N.B.); poonammehta@dsc.du.ac.in (P.M.); 2Department of Zoology, Kalindi College, University of Delhi, Delhi 110008, India; manishaarorapandit@kalindi.du.ac.in; 3Department of Zoology, Ramjas College, University of Delhi, Delhi 110007, India; cdrawat@ramjas.du.ac.in; 4Department of Botany, SGTB Khalsa College, University of Delhi, Delhi 110007, India; skaurdu@gmail.com

**Keywords:** phylloplane, microbial communities, mycoflora, *Leucaena leucocephala*

## Abstract

To address international food security concerns and sustain a growing global population, global agricultural output needs to increase by 70% by the year 2050. Current agricultural techniques to increase crop yields, specifically the application of chemicals, have resulted in a wide range of negative impacts on the environment and human health. The maintenance of good quality soil organic matter, a key concern in tropical countries such as India, requires a steady input of organic residues to maintain soil fertility. A tree with many uses, *Leucaena leucocephala*, has attracted much attention over the past decades. As per our literature review, no research has been conducted examining *Leucaena leucocephala* leaves for their fungal decomposition and their use as green manure. A study of the fungal colonization of *Leucaena leucocephala* leaves at various stages of decomposition was conducted to get an insight into which fungi play a critical role in the decomposition process. In total, fifty-two different species of fungi were isolated. There was an increase in the percentage of fungus occurrences as the leaves senesced and then finally decomposed. Almost all decomposition stages were characterized by a higher percentage occurrence of Deuteromycetes (75.47%) and by a lower rate of Ascomycetes (9.43%). A gradual increase of basidiomycetes such as unidentified sclerotia and *Rhizoctonia solani* was seen as the leaves senesced and finally decomposed. In the moist chamber, *Didymium nigripes* was the only Myxomycete isolated from completely decomposed leaves. In the present study, on average, there were more fungi in wet seasons than in the dry seasons.

## 1. Introduction

By 2050, we need to increase agricultural output by 70% to address international food security concerns and feed an exponentially growing global population. (https://sustainabledevelopment.un.org/post2015/transformingourworld, accessed on 15 March 2021). Various techniques currently in use to enhance crop yields, particularly the application of chemicals, has resulted in a wide range of negative impacts on both the environment and human health [1]. To mitigate the effects of chemical methods, mechanisms such as biological control [2] and biofertilization [3] are currently favoured. Biofertilization involves the cultivation of plants that produce green leaf manure, which aids in the betterment of the chemical, physical and biological properties of soil. In tropical countries such as India, the maintenance of good quality soil organic matter is a major concern and requires a steady input of organic residues to sustain soil fertility [4].

*Leucaena leucocephala* is a tree that has attracted a lot of attention in the past few decades due to its useful characteristics, such as a caloric value of 19.4 kJ/g and specific weight ranging from 0.50−0.59 kg/cm^3^. These properties make it conducive for paper and coal production [5,6]. It is utilised for making light structures and containers, as well as various types of fences and furniture such as tables, because it is easily machinable, porous to water-soluble preservatives and non-deformable after drying [5,7]. In addition, it is employed as a shade tree in a variety of plantations [7,8,9,10], as a mulch [11] and nitrogen fixer in the soil [12,13,14,15], as well as being used for regeneration of bare regions, slopes and pastures [9,16,17]. The leaves and seeds from *Leucaena* are also used as a food source in rural parts of Southeast Asia and Central America since they contain a high amount of fat (greater than 5.5%); particularly the fatty acids behenic, palmitic, stearic, linoleic, lignoceric and oleic acids [16,18]. It is also used as a beverage in place of coffee [12,19] and as a deworming agent [20,21]. A variety of brown, black and red pigments can be extracted from the bark, leaves and pods of *Leucaena*. On ingestion, it exhibits emmenagogic and abortive characteristics and is routinely used as a household medicine [22]. It is also beneficial for honeybees and exhibits defiance against dry climatic conditions in contrast to pastures and other forages that undergo browning and lose nutrients, thereby, ensuring availability of forage [23]. It is between 50% to 70% palatable and digestible [24] with a high nutritional range of 22−28% protein. It can, therefore, be used as a feed for both ruminants and non-ruminant livestock. Its high β-carotene and protein content along with its amino acid composition puts it on par with fodder obtained from alfalfa, fishmeal and soy meal [25]. It is composed of a few essential amino acids such as leucine, isoleucine, histidine and phenylalanine. The fodder is also a rich source of phosphorus, calcium and a number of other minerals [24,26,27,28], despite its lack of sodium [25,26]. A combination of total carbohydrates: 18.6%; starch: 1%; total oligosaccharides: 2.8%; reducing sugars: 4.2%; sucrose: 1.2% and raffinose: 0.6% has been reported by Kale in Akingbade [29].

The leaf surface habitat that harbours the wide range of both non-pathogenic and pathogenic microbes is called as the phylloplane [30,31]. Leaves become populated by an assortment of microorganisms from the moment they are formed and continue to sustain microbial populations through their lifetime. In the early stages of life, the leaf is possessed by several restricted or host specific parasites along with primary saprophytes [32]. These fungi obtain their nourishment either from the leaf itself or from the atmosphere. The fungi raid the readily decomposable sugars released from the surface of the leaf, faecal matter and honey dew from the leaf fauna—dead or decomposing parts of the leaf and healthy leaf tissue. The phylloplane fungi start disintegrating the cell walls and swiftly colonize the senescent leaf [33]. Very few fungi can be obtained from young leaves of the plants that are at seedling stage but as plants grow and leaves start to turn senescent the fungal population also exhibits exponential growth. The seed mycoflora is related to the epiphytic microflora of the above ground plant components from the time a plant grows from a seed [34]. The fact that treatment of seeds with fungicides suppresses epiphytic microbiota proves this.

Overall, two kinds of leaf colonizers are seen. The first are the airborne spores, called the “casual inhabitants”, that are there on the leaf surface simply by chance, while the second are the “resident inhabitants” that outnumber the casuals and are better adapted to the phylloplane [35,36]. According to Hudson [37], the pervasive fungi are those that are successful in establishing on the green leaves at an early stage and should be considered the primary saprophytes, while those with narrow host range are the restricted primary saprophytes.

According to available literature, no work exists till date to establish the fungal decomposition of leaves of *Leucaena leucocephala* and its use as green manure. The decomposition of organic matter of plant or animal origin is essentially a microbiological process in which fungi play an important role, so an assessment of the fungal colonization of Leucaena leucocephala leaves throughout its decomposition process was made to gain an understanding of the fungi that play a role in its decomposition. Considering that it is a perennial tree, this research may also be helpful in determining the impact of the environment on the dispersion of various fungi.

## 2. Materials and Methods

For the analysis of phylloplane of *Leucaena leucocephala* (Lam.) de Wit (var. K-8), the leaves were collected from Delhi state (28°4′ N and 77°2′ E) of India. Collections were made regularly at fortnightly intervals, throughout the year. Leaves were collected with a pair of sterilized forceps and scissors in sterile polythene bags and were brought to laboratory. Collections were made randomly from three different regions of trees (terminal, mid and lower regions of the tree).

Investigations were carried out on six categories of leaves:

Green leaves attached to the plant (GA).

Leaves beginning to yellow and senescence (YS).

Yellow and dry leaves just prior to abscission (YD) by spreading a plastic sheet under the tree and shaking it.

Yellow leaves immediately after abscission showing no sign of decomposition (YF).

Dried abscised leaves which were dark brown in colour and were partially decomposed, as indicated by disappearance of interveinal laminar region (PD).

Completely decomposed leaves which were black in colour due to almost total humification of laminar region. Care was taken to avoid collecting the soil associated with humified leaves (CD).

Phylloplane fungi were cultured in moist chambers for identification. For preparing moist chambers 3–4 filter papers were kept in Petri plates and were sterilized at 15 lbs. for fifteen minutes. These were moistened with sterile distilled water. Five leaves were placed in each plate under sterile conditions. Three replicates were maintained for each treatment. The moist chambers with leaves were incubated at 26 ± 1 °C for five days [38].

For quantitative estimation of fungi, the dilution plate method [39,40] was employed. One gram of leaves collected randomly at the six different stages were taken separately in 250 mL flasks containing 100 mL of sterile distilled water. The flasks were mechanically shaken for 1 h. The water after washing was decanted and serial dilutions were prepared and used for plating. The plating of 1:1000 dilution was found to be appropriate for colony counting. The plating was done using PDA and Czapek’s Dox Agar medium to get the maximum number of fungi. The entire procedure was done under sterile conditions. Colonies appearing in the culture plates, whether from spores or mycelia, were counted. Microscopic studies were done after 5 days of incubation.

For quantitative estimations, the number of fungal propagules were expressed as the number of colonies/g weight of leaves. It was calculated as
$$\frac{Average\;number\;of\;colonies\;per\;petriplate}{Amount\;of\;Aliquot\;used}\times \frac{Dilution\;of\;the\;aliquot}{Dry\;weight\;of\;the\;leaves}$$

For qualitative estimations, percent frequency of occurrence of various fungi and relative frequency occurrence of the fungal groups were calculated at different stages of leaf decomposition as well as during different months of the year. 

Percentage of frequency was calculated by using the formula given by Tresner et al. (1954).
$$\%\;\mathrm{Frequency}=\frac{Number\;of\;samples\;with\;fungus\;species}{Total\;number\;of\;samples}\times 100$$

Relative percent frequency of occurrence of various groups of fungi viz. Deuteromycetes, Zygomycetes, Ascomycetes, Basidiomycetes and Myxomycetes in different stages of decomposition was calculated as
$$\mathrm{Relative}\;\%\;\mathrm{Frequency}=\frac{Total\;percent\;frequency\;of\;occurrence\;of\;a\;group}{Total\;percent\;frequency\;of\;occurrence\;of\;all\;fungal\;groups}\times 100.$$

## 3. Results

In all, fifty-two different fungi were isolated using various techniques. Direct observation of the leaves showed few conidia of *Alternaria alternata*, chains of cells of *Aureobasidium pullulans* and acervuli of *Colletotrichum.* Along with these, mycelial growth could also be seen. Leaves are known to be colonized since the first unfolding of the leaves. However, only a few fungi are present at this stage. As the plant ages, there is an increase in the number of fungal colonies per gram of the leaves. Table 1, Table 2, Table 3, Table 4, Table 5 and Table 6 and Figure 1, Figure 2, Figure 3, Figure 4, Figure 5 and Figure 6 give the complete sequential changes in fungal population during different stages of plant growth as well as in different months of the year.

Average percent frequency of occurrence of various fungi showed a gradual increase as the leaves senesced and finally decomposed completely.

The saprophytic population on mature green leaves (GA) (Table 1) was comprised mostly of *Phaeoramularia graminicola*, *Fusarium lateritium*, *Aureobasidium pullulans*, *Cladosporium cladosporioides* and *Arthrinium cuspidatum*. These were followed in frequency of occurrence by *Aspergillus niger*, *Alternaria alternata* and *Penicillium chrysogenum*. There was also an occasional or rare appearance of a few species of *Aspergillus*, *Candida albicans*, *Chaetomium globosum*, *Pestalotia monorhincha*, *Epicoccum nigrum*, *Nigrospora sphaerica*, *Drechslera tetramera* and *Myrothecium roridum*. As the leaves senesced (YS) (Table 2) the percent frequency of occurrence of *Aureobasidium pullulans*, *Cladosporium cladosporioides*, *Aspergillus niger*, *A. flavus*, *Alternaria alernata*, *Drechslera tetramera*, *Aspergillus nidulans*, *Gloeosporium* sp. and *Phoma hibernica* increased, whereas *Colletotrichum falcatum*, *Gliocladium atrum*, *Cephalosporium acremonium*, *Mucor hiemalis* and *Monilia geophila* joined the microbial community at the later stages. During drying of the leaves before leaf fall (YD) (Table 3) there was not much colonization of the leaves by new fungi, but some of the pre-existing saprophytes like *Chaetomium globosum*, *Pestalotia monorhincha*, *Fusarium lateritium* and *Epicoccum nigrum* disappeared from the community and *Gloeosporium* sp. was the most frequently occurring fungus at this stage. The freshly fallen *Leucaena* leaves (YF) (Table 4) appeared to be less extensively colonized by commonly accepted phylloplane fungi than when they were attached to the tree. Some of the existing fungi still persisted there and showed an increase in their percentage frequency of occurrence viz. *A. niger*, *Fusarium equiseti*, whereas a few like *Alternaria alternata*, *Cladosporium cladosporioides*, *Phaeoramularia graminicola* and *Aureobasidium pullulans* showed a decline in their percent frequency of occurrence, which may be due to interactions among themselves.

After the leaves started decomposing (PD) (Table 5), there was a slight change in the existing mycoflora of the leaves; *Absidia repens*, *F. lateritium*, *Myrothecium roridum and Cladosporium cladosporioides* dominated the fungal flora. Other existing saprophytes like *Colletotrichum falcatum*, *Mucor hiemalis* and *Trichoderma viride* showed an increased percent frequency of occurrence. *Alternaria alternata*, *Aspergillus nidulans* and *Gloeosporium* sp., however, occurred only occasionally.

After the complete decomposition of the leaves (CD) (Table 6), the most common fungi included *Cladosporium cladosporioides*, *Colletotrichum falcatum*, species of *Fusarium*, *Myrothecium roridum* and *Phoma hibernica*. The characteristic feature was the appearance of myxomycete *Didymium nigripes*. This fungus did not appear at any other stage of decomposition. The percent frequency of occurrence of *Mucor hiemalis* increased greatly and there was occasional appearance of *Memnoniella echinate* and *Stachybotrys atra* which could not be isolated from any other stage of decomposition. Species of *Penicillium* were higher in green leaves, and as the leaves decomposed their percent frequency declined. 

The fungi isolated were classified into various fungal groups viz., Deuteromycetes, Zygomycetes, Ascomycetes, Basidiomycetes and Myxomycetes. The relative frequency of occurrence of various groups are shown in Figure 1, Figure 2, Figure 3, Figure 4, Figure 5 and Figure 6. It is evident from the figures that the relative percent frequency of occurrence of Deuteromycetes (75.47%) was highest at almost all the decomposition stages and that of Ascomycetes (9.43%) was the lowest. There was gradual increase in the occurrence of Deuteromycetes from when the leaves began to senesce up to their fall. But as leaves underwent decomposition their percent frequency of occurrence showed a decline. Corresponding to the decline in Deuteromycetes, the relative frequency of Zygomycetes increased sharply after the leaves had fallen and decomposed partially. However, after the complete decomposition of leaves, they showed a slight decrease from the partially decomposed stage. The relative percent frequency of occurrence of Ascomycetes was highest when the leaves had senesced but were still attached to the tree. Immediately after the leaves fell there was a sudden decrease in the value, which was followed by a sharp increase. Basidiomycetes, which were represented by sterile mycelium, unidentified sclerotia form and *Rhizoctonia solani*, showed a gradual increase as the leaves senesced and finally decomposed completely. Myxomycetes were represented by *Didymium nigripes*, which was isolated only from the completely decomposed leaves in the moist chamber.

Seasonal Variation of the Various Fungi Colonising Different Decomposition Stages is shown in Table 7 and Table 8.

Mainly three seasons affect the microfungal population on leaves i.e., rainy, winter and summer. The percent frequency of occurrence of various fungi in different seasons at different decomposition stages were calculated and then grouped into five categories of frequency:
1.0–20%Rare2.21–40%Occasional3.41–60%Frequent4.61–80%Common5.81–100%Very common

Green leaves which were attached on the tree (GA)—As observed in Table 7 and Table 8, a few species like *Alternaria*
*alternata*, Penicillium *chrysogenum*, *Cladosporium cladosporioides* occurred throughout the year but with varying frequencies. *Aureobasidium pullulans* and *Phoma hibernica* were most common in the rainy season, declining in frequency during the winters and being completely absent in the summers. *Penicillium chrysogenum* occurred rarely in summers.

In winters the most frequent fungus was *Cladosporium cladosporioide*, *Candida albicans*, *Cicinella muscae* and *Chaetomium globosum* occurred rarely and only during the rainy season. *Nigrosopora sphaerica* and *Arthrinium cuspidatum* were strictly seasonal and could be isolated only during the winters.

Yellow leaves which were still to the tree (YA)—At this decomposition stage, *Phaeoramularia graminicola*, which represented the most frequent fungus of green mature leaves during the rainy season, showed overall decline in its percent frequency of occurrence. *Aureobasidium pullulans*, *Alternaria alternate*, a species of *Aspergillus*, *Penicillium chrysogenum* and *Phoma hibernica* represented the most frequent flora during rainy and winter seasons. *Myrothecium roridum*, *Nigrospora sphaerica* and *Epicoccum nigrum* represented the strictly winter flora. *Cladosporium cladosporioides* was dominant in winters as well as at the beginning of summers, and *Epicoccum* showed a decline in its percent frequency of occurrence. *Aspergillus nidulans* was present in the rainy season, and only once in the month of May.

Yellow Senescent leaves prior to their fall on the ground (YD)—*Phaeoramularia graminicola* was the common fungus in this stage of decomposition also. *Aureobasidium pullulans*, *Alternaria alternata*, *Cladosporium cladosporioides* and *Penicillium chrysogenum* frequently occurred in all the seasons of the year. *Nigrospora* and *Arthrinium* represented the dominant flora of winter only. The frequency of *Aspergillus nidulans* increased during summers. *Colletotrichum falcatum* could be isolated only occasionally or rarely in the rainy season.

Yellow Leaves which were recently fallen on the ground (YF)—This decompositional stage of leaves shows the predominance of *Aspergillus luchuensis*, *Penicillium chrysogenum* and *Phaeoramularia* during the rainy season. *Gloeosporium* sp., which represented the most frequent fungus in this stage, could be isolated throughout the year. Some of the dominant fungi of winters like *Arthrinium cuspidatum* and *Nigrospora sphaerica*, however, occurred very rarely in this decomposition stage. *Cladosporium cladosporioides* and *Aureobasidium pullulans* were the dominant fungi in winters.

Partially Decomposed Leaves (PD)—*Colletotrichum falcatum* was the dominant fungus at this stage in the rainy season, with a few species of *Aspergillus. Cladosporium cladosporioides*, along with *Aureobasidium pullulans* and *Gloesporium* sp., were among the most frequent fungi in winter flora. *Alternaria alternata* showed decreased percent frequency of occurrence throughout. In summers, *Myrothecium roridum* and *Absidia repens* were among the most dominant fungi, with *Monilia geophila* occurring rarely.

Completely Decomposed Leaves (CD)—*Colletotrichum falcatum*, *Didymium nigripes* and *Myrothecium roridum* were the most frequent fungi of rainy season. In winters *Cladosporium cladosporioides* was the dominant fungus at this stage also. *Mucor hiemalis* and *Rhizopus stolonifer* showed greatly enhanced percent frequency of occurrence, the maximum of which could be isolated in winters. *Aspergillus pullulans* was not isolated from leaves of this stage at all. *Alternaria alternata* could also be isolated only rarely. However, occasional occurrence of *Memnoniella echinata* and *Stachybotrys atra* could be seen during the winters. The summer fungal flora showed the predominance of *Cladosporium cladosporioides*, *Myrothecium roridium*, *Phoma hibernica* and *Colletotrichum falcatum*.

## 4. Discussion

During the present study, the fungi were isolated from leaves at six different decomposition stages in different seasons of the year. It is evident from the data that by the time the leaves of Leucaena reach the surface litter, they are substantially colonized by a variety of parasitic and saprophytic fungi. This contrasts with Hering’s [41] conclusion that the fungi dominant on fallen oakwood litter were not present to any great extent before leaf fall. However, leaves of *Quercus rotundifolia Lam* [42], Pinus sylvestris [43,44], Populus tremuloides [45], Ilex aquifolium [46], Sesamum orientale, Gossypium hirsutum [40] and Mangifera indica [47] have colonization of leaves by fungi before their fall.

According to Hudson’s scheme [37], the primary saprophytic colonizers are represented by Ascomycetes and Deuteromycetes, of which Cladosporium herbarum, Alternaria alternate, Epicoccum nigrum, Aureobasidium pullulans and Botrytis cinerea are common components of successions. These are present on the phylloplane of living leaves as spores, which become vegetatively active only at leaf senescence. Most of these common fungi were isolated from leaves of Leucaena also. However, in the present investigation, colonization of leaves by these fungi occurred at very early stage, as was also reported by Dickinson [48], Lindsey and Pugh [49], Mishra and Dickinson [46] Promputtha, et al. [50] and Wildman and Parkinson [45]. Therefore, commencement of pathogenic activity did not coincide with leaf senescence. Initially, at the seedling stage, there were very few fungal propagules on the leaves. As the plant aged, the number of fungal colonies per gram of leaves increased, which may be due to prolonged exposure of the leaves to the air spores, and it is also believed that ageing leaves produce a greater number of exudates which enhance fungal colonization. As the leaves senesced, there was a gradual increase in internal colonization of the leaves, indicating a slow penetration of the fungi into leaves with time. Heavy colonization of senescing leaves has also been reported by several other works [40,46,49,50,51,52].

After the leaves fell on the ground and their decomposition was initiated, the primary saprophytes on leaves were joined by new colonizers and a few pre-existing fungi showed increased percentage frequency of occurrence. Some of the new colonists included Absidia repens, Bipolaris sp. Stachybotrys atra, Memnoniella echinata, Mucor hiemalis and Didymium nigripes, which were present on litter. The pre-existing species like Pestalotia monorhincha, Epicoccum nigrum and Fusarium lateritium could not be isolated from the litter.

The persistence of primary saprophytes during the initial decay period has been attributed to various factors. The period of the parasitic phase, which enables them to penetrate and establish in freshly decaying tissue, is very short. According to Hogg [53] and Visser and Parkinson [54] most of the primary saprophytes have a high rate of sporulation, and due to their ability to survive under drought conditions, they persist for a longer time. Higher sporulation in the fungi was probably due to the availability of a large amount of soluble nutrients.

Extensive work has been done on various aspects of the microbial ecology of leaf and litter in relation to the degradation and fertility of soil [4,50,55,56,57]. However, much is not known about the decomposition potentials of the fungal colonizers. Sharma and Mukerji [58,59] recognized four different types of fungal colonization patterns on leaves of Sesamum orientale and Gossypium hirsutum. Differences were determined chiefly by the developmental stage of the organ on which the organism occurred, the substrate it provided, the capacity of the fungus to utilize the available substrates, and the potential of the fungus to degrade the organ following senescence.

In the current study, Alternaria alternata, Cladosporium cladosporioides, different species of *Fusarium*, Aspergillus, Penicillium chrysogenum, Trichoderma viride and Phoma hibernica appeared to have no restricted distribution. They occurred significantly during the living as well as the senescent and decaying phases of the leaves. Although Penicilli, Aspergilli and *Trichoderma viride* occurred quite significantly in dilution plates, these were absent or rare in moist chambers. Dickinson [60] and Webster [61] classified Aspergilli and Penicilli as casual inhabitants which play negligible roles in decomposition. Sharma et al. [40] could isolate Trichoderma viride very rarely, and they grouped it under non-decomposers, but Hering [41] showed that Penicillium sp. And Trichoderma viride, though isolated only in dilution plates, did have a role in decomposition, and are important cellulose degrading fungi.

The completely decomposed leaves incubated in moist chambers were peculiar in that they were exclusively colonized by the sporulating stages of Didymium nigripes. This agrees with earlier reports on cotton leaves and on leaves and stems of Gossypium and *Sesamum* [62]. The plating of surface washings of decaying organs exhibited a high number of imperfect fungi, especially those which have been considered decomposers [58].

Another interesting observation was the frequent isolation of Mucorales from the surface washing of decaying leaves. The frequency of Mucorales significantly increased on highly decomposed leaves. Similar results were obtained by Sharma and Mukerji [62]. However, Sharma et al. [40] has not mentioned the position of yeasts like fungi viz. Aureobasidium pullulans in a successional pattern. Phaeoramularia graminicola was also isolated very frequently in the present study on attached leaves. With leaf fall, there was a decline in frequency of occurrence of Aureobasidium pullulans. This decline in yeast and Aureobasidium frequency may have been due to the dryness of the fallen leaves. Ruscoe [44] used a direct observation technique and showed that Aureobasidium colonized the phylloplane of young Nothofagus leaves. This fungus formed vigorously, growing colonies whose development increased with increasing leaf age and then declined at leaf fall. However, autolysis of fungal hyphae was not reported by Wildman and Parkinson [45].

Garrett [63] proposed a generalized fungal sequence on plant material within or on the soil. The sequence is as follows: weak parasites → primary saprophytic sugar fungi → cellulose decomposers → secondary sugar fungi → lignin decomposers + associated fungi.

The colonization pattern of Leucaena leaves at different decomposition stages followed the general trend. The green mature leaves were colonized predominantly by Deuteromycetes, along with the rare appearance of other groups of fungi. As the leaves senesced and decomposed, the soil Ascomycetes (which are known to be important cellulose degraders) increased in their percent frequency of occurrence. Litter in its final stages of decomposition showed enhanced growth of Mucorales and Basidiomycetes. The Myxomycetes isolated in the present study and reported earlier [62,64] from the completely decomposed leaf litter, however, represent the secondary saprophytes of the Hudson’s scheme [37].

Since *Leucaena* is an evergreen tree, all kinds of attached and fallen leaves were present at the same time. It may be that some fungi are found in all stages of leaf decomposition, suggesting that there is a two-way relationship between the mycoflora of living and dead leaves. These will affect successful establishment, growth, reproduction and survival of both phylloplane inhabitants and litter decomposers.

Environmental factors play a very important role in quantitative and qualitative distribution of the fungi on the leaves. In the present study, average number of fungi were more in wet season than in the dry season. The greater number of fungal taxa recorded in wet season is like the observations of Almaguer, et al. [65,66], Jothish & Nayar [67] and Stennett & Beggs [68]. This may be due to high relative humidity, moderate temperature and lower sunshine duration. According to Diem [69], in the rainy season the cuticle is constantly wet, which is suitable for the growth of fungi. Irrespective of plant age, several fungi occurred only during a specific period of investigation. For example, Phaeoramularia graminicola *and* Didymium nigripes were strictly isolated in the rainy season. Arthrinium cuspidatum, Nigrospora sphaerica, Epicoccum nigrum and Pestalotia monorhincha were observed only in the winters. It is thus evident that microbial activity depends on the micro and macro environmental conditions and the substrate characteristics. Studies on the associated fungi of the same plant grown in different localities, therefore, is of great interest.

## 5. Conclusions

Average percent frequency of occurrence of various fungi gradually increased as the leaves senesced and finally decomposed completely. The mature green leaves were colonized by *Chaetomium golobosum*, *Pestalotia monorhincha*, *Fusarium lateritium* and *Epicoccum nigrum* along with *Phaeoramularia graminicola*, *Aureobasidium pullulans*, *Alternaria alternate* and *Cladosporium* sp., which were very frequent. As the leaves senesced, the percentage frequency of occurrence of a few pre-existing fungi increased, whereas Colletotrichum falcatum, Gliocladium atrum, Cephalosporium acremonium, Mucor hiemalis and Monilia geophila later joined the microbial community. Upon drying the leaves before leaf fall there was not much colonization of the leaves by new fungi, but some of the pre-existing fungi like *Chaetomium globosum*, *Pestalotia monorhinch*, *F. lateritium* and *Epicoccum nigrum* disappeared from the community and Gloeosporium appeared, which was the most frequently occurring fungus at this stage. The freshly fallen *Leucaena* leaves appeared to be less extensively colonized by commonly accepted phylloplane fungi than when they were on the tree. After the leaves started decomposing there was slight change in the existing mycoflora of the leaves. *Absidia*, *Fusarium lateritium*, *Myrothecium roridum* and *Cladosporium* sp. dominated the fungal flora. After complete decomposition of the leaves, the characteristic feature was the appearance of Myxomycetes *Didymium nigripes*. *Memnoniella echinata* and *Stachybotrys atra* could also be isolated only from this stage of decomposition. In the present study, on average, the number of fungi was greater in the wet season than in the dry season.

## Figures and Tables

**Figure 1 jof-08-00608-f001:**
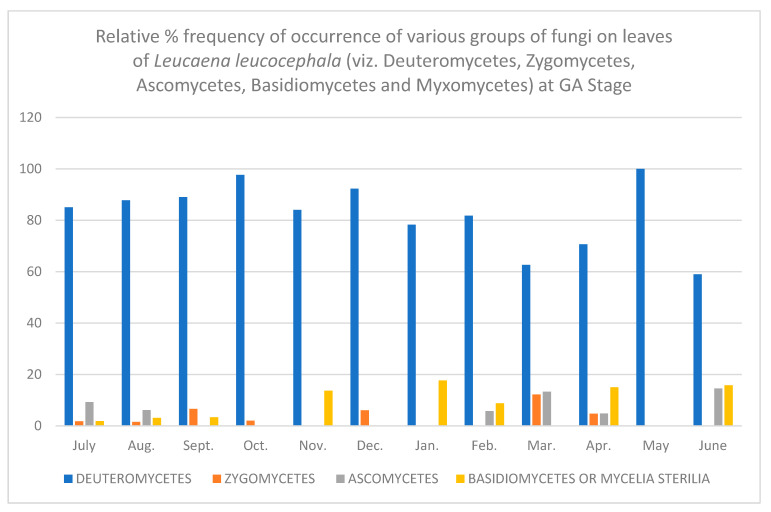
Relative % frequency of occurrence of various groups of fungi on leaves of *Leucaena leucocephala* (viz. Deuteromycetes, Zygomycetes, Ascomycetes, Basidiomycetes and Myxomycetes) at GA Stage.

**Figure 2 jof-08-00608-f002:**
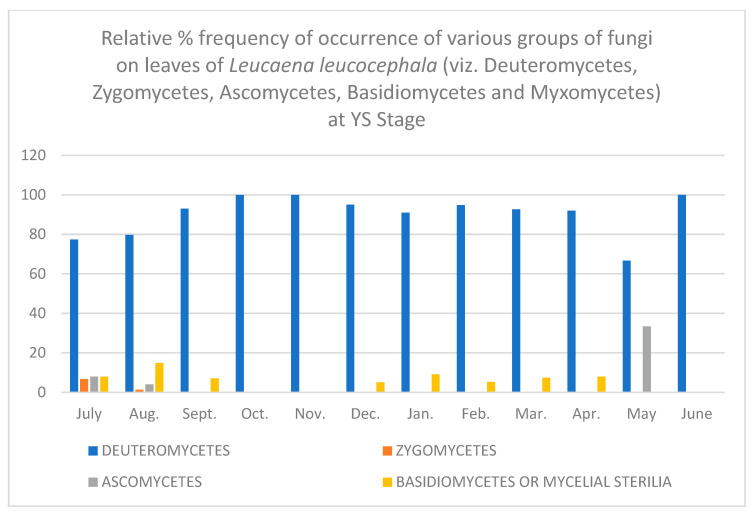
Relative % frequency of occurrence of various groups of fungi on leaves of *Leucaena leucocephala* (viz. Deuteromycetes, Zygomycetes, Ascomycetes, Basidiomycetes and Myxomycetes) at YS Stage.

**Figure 3 jof-08-00608-f003:**
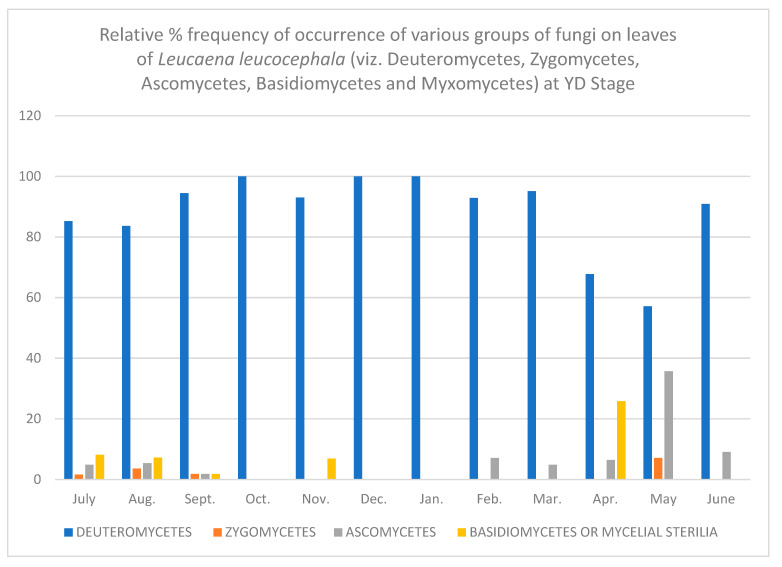
Relative % frequency of occurrence of various groups of fungi on leaves of *Leucaena leucocephala* (viz. Deuteromycetes, Zygomycetes, Ascomycetes, Basidiomycetes and Myxomycetes) at YD Stage.

**Figure 4 jof-08-00608-f004:**
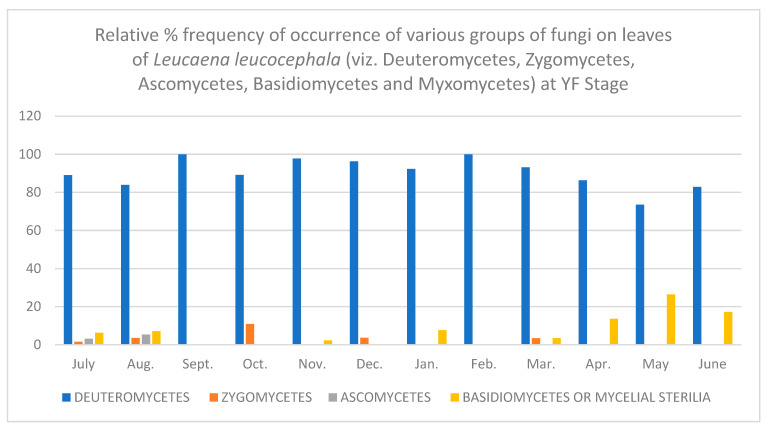
Relative % frequency of occurrence of various groups of fungi on leaves of *Leucaena leucocephala* (viz. Deuteromycetes, Zygomycetes, Ascomycetes, Basidiomycetes and Myxomycetes) at YF Stage.

**Figure 5 jof-08-00608-f005:**
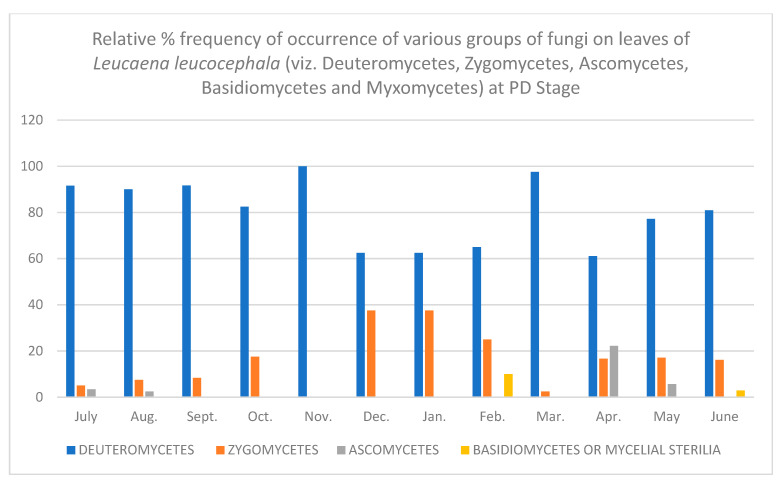
Relative % frequency of occurrence of various groups of fungi on leaves of *Leucaena leucocephala* (viz. Deuteromycetes, Zygomycetes, Ascomycetes, Basidiomycetes and Myxomycetes) at PD Stage.

**Figure 6 jof-08-00608-f006:**
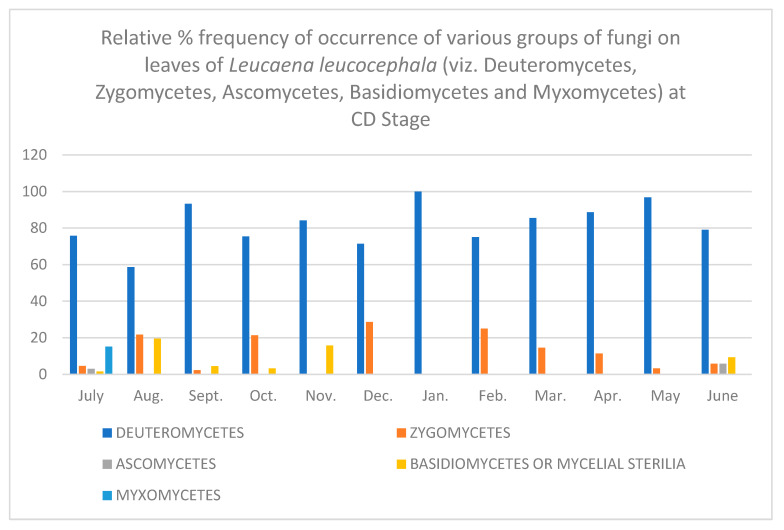
Relative % frequency of occurrence of various groups of fungi on leaves of *Leucaena leucocephala* (viz. Deuteromycetes, Zygomycetes, Ascomycetes, Basidiomycetes and Myxomycetes) at CD Stage.

**Table 1 jof-08-00608-t001:** Percentage frequency of occurrence of various fungi on Green Attached Leaves (GA) of *Leucaena leucocephala*.

	Months of the Year											
Fungi	July	Aug.	Sept.	Oct.	Nov.	Dec.	Jan.	Feb.	Mar.	Apr.	May	June
** *DEUTEROMYCETES* **												
*Alternaria alternata*	90	80	50	30	40	60	50	60	20	100	30	
*Arthrinium cuspidatum*								60				
*Aspergillus flavus*	30	20	50	10	10					40		
*A. fumigatus*	20		10							50		
*A. humicola*										20		
*A. luchuensis*	30	20										30
*A. niger*	80	80	50	50	10		50	80	20		50	20
*A. sulphureus*	20	30		10	30							20
*Aureobasidium pullulans*	90		100	40	90	100	70	30	10	20		
*Candida albicans*	10											
*Cladosporium cladosporioides*		10	80	100	100	80	60	60				
*Curvularia lunata*	20	40	80		20							
*C. pallescens*				20								
*Drechslera tetramera*	20	10	10	20								
*Drechslera hawaiiensis*				10								
*Epicoccum nigrum*				10	20	30	50					
*Fusarium lateritium*											100	
*F. oxysporum*		20	50	10	10					30		60
*Glaeosporium* sp.		10	50	20								
*Myrothecium roridum*			10		20					30		
*Nigrospora sphaerica*				60	10	40						
*Penicillium chrysogenum*	30	90	90	80					20	10		
*P. citrinum*									20			
*P. funiculosum*		40		10					20			
*Pestalotia monorhincha*			30		30							
*Phaeoramularia graminicola*		80	100									
*Phoma hibernica*	30	50	50	20	40							
*Trichoderma viride*				10	10							
** *ZYGOMYCETES* **												
*Choanephora cucurbitarum*	10		50									
*Circinella muscae*			10									
*Rhizopus stolonifer*		10		10		20			20	20		
** *ASCOMYCETES* **												
*Chaetomium globosum*		10										
*Aspergillus nidulans*	50	30						20	20	20		30
** *BASIDIOMYCETES OR MYCELIA STERILIA* **												
Sterile mycelium	10	20	30		70		60	30		50		30
*Unidentified Sclerotial form*										10		

**Table 2 jof-08-00608-t002:** Percentage frequency of occurrence of various fungi on yellow attached Leaves (YS). *Leucaena leucocephala*.

	Months of the Year											
Fungi	July	Aug.	Sept.	Oct.	Nov.	Dec.	Jan.	Feb.	Mar.	Apr.	May	June
** *DEUTEROMYCETES* **												
*Alternaria alternata*	60	80	20		50	60			70	70	60	
*Arthrinium cuspidatum*									20			
*Aspergillus flavus*	90	50	20							10		
*A. fumigatus*	50									20		
*A. luchuensis*	100	30										20
*A. niger*	90	80	80	10						30	20	50
*A. sulphureus*											20	
*A. terreus*	10											
*Aureobasidium pullulans*		50	80		100	60						
*Cephalosporium reseogriseum*				10								
*Cladosporium cladosporioides*			60		100	100	100	90	100	80		
*Colletotrichum falcatum*				40	20							
*Curvularia lunata*	30											
*Drechslera tetramera*	30	60										
*Epicoccum nigrum*						20			20			
*Fusarium oxysporum*	60		40		20					10		
*F. semitectum*	10					20						30
*Gloeosporium* sp.			40	80	50	20						
*Gliocladium atrum*			40									
*Myrothecium roridum*					50							
*Nigrospora sphaerica*					100	80	80	60	20			
*Penicillium chruysogenum*	30	90	60	20								
*P. citrinum*					20					10		
*P. funiculosum*	10											10
*Phaeoramularia graminicola*		80	20									
*Phoma hibernica*		60	60		100	20	20	30	20			
*Trichoderma viride*	10	10	10									
** *ZYGOMYCETES* **												
*Circinella muscae*	20											
*Mucor hiemalis*	10											
*Rhizopus stolonifer*	20	10										
** *ASCOMYCETES* **												
*Aspergillus nidulans*	60	30									50	
** *BASIDIOMYCETES OR MYCELIAL STERILIA* **												
Sterile mycelium	60	60	40			20	20	10	20	20		
Unidentified Sclerotial form		50										

**Table 3 jof-08-00608-t003:** Percentage frequency of occurrence of various fungi on leaves before they fall on ground (YD).

	Months of the Year											
Fungi	July	Aug.	Sept.	Oct.	Nov.	Dec.	Jan.	Feb.	Mar.	Apr.	May	June
** *DEUTEROMYCETES* **												
*Alternaria alternata*	10	50	10		10	100	100	90	100	90	20	10
*Arthrinium cuspidatum*					30	20						
*Aspergillus flavus*	100	40	40	90						30	20	30‘
*A. fumigatus*	70											
*A. luchuensis*	70			20								10
*A. niger*	100	80	50	80						30	30	40
*A. sulphureus*										30		
*A. terreus*	10											
*Aureobasidium pullulans*	20	10		10	100	60		60	100			
*Cladosporium cladosporioides*			50	100	100	100	80	90	80		10	
*Colletotrichum falcatum*		20		40								
*Curvularia lunata*	20	30	10	20						30		
*Cylindrocarpon* sp.	10											
*Drechslera tetramera*		10										
*Epicoccum nigrum*						20			80			
*Fusarium oxysporum*	60	10	50	20	10							
*Gloeosporium* sp.			100									
*Monilia geophila*		10		10	20							
*Myrothecium roridum*		10			20							
*Nigrospora sphaerica*			10	80	20	60			10			
*Penicillium chrysogenum*	30	80	90	70			30	20	20			
*P. funiculosum*	10	40		10								10
*Phaeoramularia graminicola*		60	90									
*Phoma hibernica*		10		10	90	20						
*Trichoderma viride*	10		10									
** *ZYGOMYCETES* **												
*Choanephora cucurbitarum*		20										
*Rhizopus stolonifer*	10		10								10	
** *ASCOMYCETES* **												
*Aspergillus nidulans*	30	30	10					20	20	20	50	10
** *BASIDIOMYCETES OR MYCELIA STERILIA* **												
Sterile mycelium	50	40	10		30					50		
Unidentified sclerotial form										30		

**Table 4 jof-08-00608-t004:** Percentage frequency of occurrence of various fungi on yellow-Fallen leaves (YF).

	Months of the Year											
Fungi	July	Aug.	Sept.	Oct.	Nov.	Dec.	Jan.	Feb.	Mar.	Apr.	May	June
** *DEUTEROMYCETES* **												
*Alternaria alternate*	40	50		10	20					40	30	
*Arthrinium cuspidatum*					10							
*Aspergillus flavus*	60	40		50						10	10	30
*A. fumigatus*	70									40		70
*A*. *humicola*										10		
*A. luchuensis*	100											20
*A. niger*	100	80	60	70						20	50	60
*A. sulphureus*	10			10						20		50
*A. terreus*				10								
*Aureobasidium pullulans*	10	10		10	100	100	90	70	10	10		
*Choanephora cucurbitarum*		20										
*Cladosporium cladosporioides*			20	100	100	100	90	60	10	20	30	
*Colletotrichum falcatum*		20	60	20	40				50			20
*Curvularia lunata*	10	30	40									10
*C. pallescens*				10								
*Cylindrocarpon* sp.	40	10										
*Drechslera tetramera*	30											
*Epicoccum nigrum*				20								
*Fusarium equiseti*	60								80			
*F. lateritium*											90	10
*F. oxysporum*		10	20							10		
*F. semitectum*				10	20					80		
*Gloesporium* sp.			100		100	60	60	70	80			
*Mucor hiemalis*				20					10	20	10	
*Myrotehcium roridum*		10								100	20	20
*Nigrospora sphaerica*					20							
*Penicillium chrysogenum*	10	80	100	40								
*P. funiculosum*	10	40		10								
*P. nigricans*	20											
*Phaeoramularia grominicola*		60	80									
*Phoma hibernica*		10	10		20				30			
*Trichoderma viride*			10	20							10	
** *ZYGOMYCETES* **												
*Choanephora cucurbitarum*		20										
*Rhizopus stolonifer*	10			50		10			10			
** *ASCOMYCETES* **												
*Aspergillus nidulans*	20	30										
** *BASIDIOMYCETES OR MYCELIA STERILIA* **												
Sterile mycelium	40	40			10		20			60	50	50
*Rhizoctonia solani*									10		40	10

**Table 5 jof-08-00608-t005:** Percentage frequency of occurrence of various fungi on partially decomposed leaves (PD).

	Months of the Year											
Fungi	July	Aug.	Sept.	Oct.	Nov.	Dec.	Jan.	Feb.	Mar.	Apr.	May	June
** *DEUTEROMYCETES* **												
*Alternaria alternata*		10		10	20				70			
*Aspergillus flavus*	100	90	10	80						20	60	50
*A. fumigatus*	90	20										100
*A. luchuensis*	100	10		20								90
*A. niger*	100	100	40	100	20		20	30		80	90	100
*A. sulphureus*												30
*A. terreus*										20		
*Aureobasidium pullulans*				80	40				50			10
*Cladosporium cladosporioides*				80	100	100	80	80	80	40		
*Colletotrichum falcatum*	100	80		10					30			
*Curvularia lunata*				10						20		
*Cylindrocarpon* sp.		20										
*Drechslera tetramera*	10											
*Fusarium lateritium*											100	80
*F. oxysporum*	30			20	80							
*F. semitectum*								20	20	20		
*Gloeosporium* sp.		10	20		60							
*Monilia geophila*									60		20	
*Myrothecium roridum*												90
*Nigrospora sphaerica*									10			
*P. chrysogenum*		10	40	10					40			
*Penicillium citrinum*									40			
*P. funiculosum*		10										
*Phoma hibernica*				10	60							
*Trichoderma viride*	10			40						20		
** *ZYGOMYCETES* **												
*Absidia repens*												100
*Choanephora cucurbitarum*	10	20	10									
*Mucor hiemalis*		10		100		60	60	40	10	40	50	10
*Rhizopus stolonifer*	20							10		20	10	
** *ASCOMYCETES* **												
*Aspergillus nidulans*	20	10								80	20	
** *BASIDIOMYCETES OR MYCELIA STERILIA* **												
*Sterile mycelium*								20				20

**Table 6 jof-08-00608-t006:** Percentage frequency of occurrence of various fungi on completely decomposed leaves (CD).

	Months of the Year											
Fungi	July	Aug.	Sept.	Oct.	Nov.	Dec.	Jan.	Feb.	Mar.	Apr.	May	June
** *DEUTEROMYCETES* **												
*Alternaria alternata*						40				20		
*Aspergillus flavus*	90	90	50	90	80				100	100	60	70
*A. fumigatus*	40	10							10			100
*A. luchuensis*	20	20	30	30						10		80
*A. niger*	90	100	100	100				50	40	100	40	100
*A. parasiticus*									10			
*A. sulphureus*				90						40		100
*A. terreus*									10	40		20
*Bipolaris* sp.									10			
*Cladosporium cladosporioides*				100	80	100	80	100	100			
*Colletotrichum falcatum*	70		100						90		100	80
*Fusarium lateritium*											100	
*F. oxysporum*	90		40						50			
*F. semitectum*		50		20		60			50	70		40
*Memnoniella echinata*						40						
*Myrothecium roridum*	100											
*Penicillium chrysogenum*			50									90
*P. citrinum*									30			
*Phaeoramularia graminicola*			10									
*Phoma hibernica*									90			
*Stachybotrys atra*						10	10					
*Trichoderma virde*			40	30						10		
** *ZYGOMYCETES* **												
*Absidia repens*												40
*Choanephora cucurbitarum*		30	10									
*Mucor hiemalis*	20	30		100		40		50	40	50	10	
*Rhizopus stolonifer*	10	40		30		60			60			10
** *ASCOMYCETES* **												
*Aspergillus nidulans*	20											50
** *BASIDIOMYCETES OR MYCELIA STERILIA* **												
Sterile mycelium	10	90	20	20	30							80
** *MYXOMYCETES* **												
*Didymium nigripes*	100											

**Table 7 jof-08-00608-t007:** Average percentage frequency of occurrence of various fungi at different decomposition stages of the leaves in different seasons.

Fungi	Green Attached (GA)			Yellow Attached (YA)			Yellow before Fall (YF)			Recently Fallen (RF)			Partially Decomposed (PD)			Completely Decomposed (CD)		
	R	W	S	R	W	S	R	W	S	R	W	S	R	W	S	R	W	S
*Abisidia repens*	–	–	–	–	–	–	–	–	–	–	–	–	–	–	100	–	–	–
*Alternaria alternata*	73	48	50	53	55	67	23	80	55	45	15	35	10	15	70	–	40	20
*Arthrinium cuspidatum*	–	60	–	–	–	20	–	25	–	–	10	–	–	–	–	–	–	–
*Aspergillus flavus*	33	10	40	53	–	25	60	90	27	33	50	27	67	80	43	77	85	83
*A. fumigatus*	15	–	50	50	–	20	70	–	–	70	–	27	55	–	100	25	–	55
*A. humicola*	–	–	20	–	–	–	–	–	–	–	–	10	–	–	–	–	–	–
*A. luchuensis*	25	–	30	65	–	20	70	20	10	100	–	20	55	–	90	23	30	45
*A. niger*	70	48	30	83	10	33	77	80	33	80	70	43	80	43	90	97	75	70
*A. parasiticus*	–	–	–	–	–	–	–	–	–	–	–	–	–	–	–	–	–	10
*A. sulphureus*	25	20	20	–	–	20	–	–	30	10	10	35	–	–	30	–	90	70
*A. terreus*	–	–	–	10	–	–	10	–	–	–	10	–	–	–	20	–	–	23
*Aureobasidium pullulans*	95	66	15	65	80	–	15	58	100	10	74	10	–	60	30	–	–	–
*Basidiomycete* (Sclerotial)	–	–	10	50	–	–	–	–	30	–	–	–	–	–	–	–	–	–
*Bipolaris* sp.	–	–	–	–	–	–	–	–	–	–	–	–	–	–	–	–	–	10
*Candida albicans*	10	–	–	–	–	–	–	–	–	–	–	–	–	–	–	–	–	–
*Cephalosporium roseogriseum*	–	–	–	–	10	–	–	–	–	–	–	–	–	–	–	–	–	–
*Chaetomium globosum*	10	–	–	–	–	–	–	–	–	–	–	–	–	–	–	–	–	–
*Choanephora cucurbitarum*	30	–	–	–	–	–	20	–	10	20	–	–	13	–	–	20	–	–
*Circinella muscae*	10	–	–	20	–	–	–	–	–	–	–	–	–	–	–	–	–	–
*Cladosporium cladosporioides*	10	84	60	60	98	90	50	94	45	20	90	20	–	88	60	–	92	100
*Colletotrichum falcatum*	–	–	–	–	30	–	20	40	–	40	30	35	90	10	30	85	–	90
*Curvularia lunata*	47	20	–	30	–	–	30	20	30	27	–	10	–	10	20	–	–	–
*C. pallescens*	–	20	–	–	–	–	–	–	–	–	10	–	–	–	–	–	–	–
*Cylindrocarpon* Sp.	–	–	–	–	–	–	10	–	–	25	–	–	20	–	–	–	–	–
*Didymium nigripes*	–	–	–	–	–	–	–	–	–	–	–	–	–	–	–	100	–	–
*Drechslera tetramera*	13	20	–	45	–	–	10	–	30	30	–	30	10	–	–	–	–	–
*D. hawaiiensis*	10	–	–	–	–	–	–	–	–	–	–	–	–	–	–	–	–	–
*Aspergillus nidulans*	40	20	20	45	–	50	35	20	50	’25	–	–	15	–	50	20	–	50
*Epiococcum*	–	28	–	–	20	20	–	20	80	–	20	–	–	–	–	–	–	–
*Fusarium equiseti*	–	–	–	–	–	–	–	–	–	60	–	80	–	–	–	–	–	–
*F. lateritium*	–	–	100	–	–	–	–	–	–	–	–	50	–	–	90	–	–	100
*F. oxysporum*	35	10	45	50	20	10	40	15	–	15	–	10	30	50	–	65	–	50
*F. semitectum*	–	–	–	10	20	30	–	–	–	–	15	80	–	20	20	50	40	53
*Gliocladium atrum*	–	–	–	40	–	–	–	–	–	–	–	–	–	–	–	–	–	–
*Gloeosporium* Sp.	30	20	–	40	50	–	100	–	–	100	73	80	15	60	–	–	–	–
*Memnoniella echinata*	–	–	–	–	–	–	–	–	–	–	–	–	–	–	–	–	40	–
*Monilia geophila*	–	–	–	–	–	–	–	10	–	–	–	–	–	–	40	–	–	–
*Mucor hiemalis*	–	–	–	10	–	–	–	–	–	–	20	33	10	65	40	25	63	33
*Myrothecium roridum*	10	20	30	–	50	–	10	20	–	10	–	47	–	–	90	100	–	90
*Nigrospora sphaerica*	0	37	–	–	80	20	10	53	10	–	20	–	–	–	10	–	–	–
*Penicillium chrysogenum*	70	80	15	60	20	–	67	40	20	63	40	–	25	10	–	50	–	–
*P. citrinum*	–	10	20	–	20	10	–	–	–	–	–	–	–	–	40	–	–	30
*P. funiculosum*	40	–	20	10	–	10	25	10	10	25	10	–	10	–	–	–	–	–
*P. nigricans*	–	–	–	–	–	–	–	–	–	20	–	–	–	–	–	–	–	–
*Pestalotia monorhincha*	30	30	–	–	–	–	–	–	–	–	–	–	–	–	–	–	–	–
*Phaeoramularia graminicola*	90	–	–	50	–	–	75	–	–	70	–	–	–	–	–	10	–	–
*Phoma hibernica*	65	30	–	60	43	20	10	40	–	10	20	30	–	35	–	–	–	90
*Rhizoctonia solani*	–	–	–	–	–	–	–	–	–	–	–	20	–	–	–	–	–	–
*Rhizopus stolonifer*	10	15	20	10	–	10	10	–	10	10	30	10	20	10	15	25	45	35
*Stachybotrys atra*	–	–	–	–	–	–	–	–	–	–	–	–	–	–	–	–	10	–
Sterile mycelium	20	53	40	53	18	20	33	30	50	40	20	53	–	20	20	–	27	80
*Trichoderma viride*	–	10	–	10	–	–	10	–	–	10	20	10	10	40	20	–	30	10

**Table 8 jof-08-00608-t008:** Seasonal variation of various fungi at different decomposition stages of the leaves.

Different Decomposition Stages	Season	Very Common	Common	Frequent	Occasional	Rare
Green Leaves Which Were Attached on the Tree (GA)	Rainy Season	*Aureobasidium pullulans*, *Phaeoramularia graminicola*	*Aspergillus niger*, *Alternaria alternate*, *Penicillium chrysogenum*, *Phoma hibernica*	*Curvularia lunata*	*Aspergillus flavus*, *A. luchuensis*, *A. sulphureus*, *Aspergillus nidulans*, *Penicillium funiculosum*, *Fusarium oxysporum*, *Gloeosporium* sp., *Pestalotia monorhincha.*	*Candida albicans*, *Chaetomium globosum*, *Circinella muscae*, *Cladosporium* sp., *Drechslera tetramera*, *D. hawiiensis*, *Myrothecium roridum*, *Rhizopus stolonifer*, *sterile mycelium.*
	Winter Season	*Cladosporium cladosporioides*	*Aureobasidium pullulans*, *P. chrysogenum*	*A. niger*, *Alternaria alternate*, *Arthrinium cuspidatum*, *Sterile mycelium.*	*Epicoccum nigrum*, *Nigrospora sphaerica*, *Pestalotia monorhinca*, *Phoma hibernica.*	*Aspergillus flavus*, *Curvularia lunata*, *C. pallescens*, *Drechslera tetramera*, *Aspergillus nidulans*, *F. oxysporum*, *Gloeosporium* sp. *Myrothecium roridum*, *P. citrinum*, *Rhizopus stolonifer*, *Trichoderna viride.*
	Summer Season	*Fusarium lateritium*	*None*	*Aspergillus fumigatus*, *Alternaria alternate*, *Cladosporium cladosporioide*, *Fusarium oxysporum*	*Aspergillus flavus*, *A. luchuensis*, *A. niger*, *sterile mycelium.*	*Aspergillus humicola*, *A. sulphureus*, *Aspergillus nidulans*, *Penicillium chrysogenum*, *P. citrinum*, *Rhizopus stolonifer*, *unidentified Basidomycetes*
Yellow Leaves which were still attached on the tree (YA)	Rainy Season	*Aspergillus niger*	*Aspeurgillus luchensis*, *Aureobasidium pullulans*	*Aspergillus flavus*, *A. fumigatus*, *Alternaria alternate*, *Cladosporium cladosporioides*, *Drechslera tetramera*, *Aspergillus nidulans*, *Fusarium oxysporum Penicillium chrysogenum*, *Phaeoramularia graminicola*, *Phoma hibernica*, *sterile mycelium*, *unidentified Basidiomycete*	*Curvularia lunata*, *Gloesporium* sp., *Gliocladium atrum*	*Aspergillus terreus*, *Circinella muscae*, *Fusarium semitectum*, *Mucor hiemalis*, *Penicillium funiculosum*, *Rhizopus stolonifer*, *Trichoderma viride*
	Winter Season	*Cladosporium cladosporioides*	*Aureobasidium pullulans*, *Nigrospora sphaerica*	*Alternaria alternate*, *Gloeosporium* sp., *Myrothecium roridum*, *Phoma hibernica*	*Collectotrichum falcatum*	*Aspergillus niger*, *Cephalosporium roseogriseum*, *Epicoccum nigrum*, *Fusarium oxysporum*, *F. semitectum*, *Penicillium chrysogenum*, *P. citrinum*, *sterile mycelium.*
	Summer Season	*Cladsoporium cladosporioides*	*Alternaria alternate*	*Aspergillus nidulans*	*Aspergillus flavus*, *A. niger*, *Fusarium semitectum*	*Aspergillus fumigatus*, *A. luchensis*, *A. sulphureus*, *Arthrinium cuspidatum*, *Epicoccum nigrum*, *Fusarium oxysporum Nigrospora sphaerica*, *penicillium citrinum*, *P. funiculosum Rehizopus nigricans*, *sterile mycelium.*
Yellow Senescent Leaves prior to their fall on the ground (YD)	Rainy Season	*Gloeosporium* sp.	*Aspergillus fumigatus*, *A. luchuensis*, *A. niger*, *Penicillium chrysogenum*, *Phaeoramularia graminicola*	*Aspergillus flavus*, *Cladosporium cladosporioides*	*Alternaria alternate*, *Curvularia lunata*, *Aspergillus nidulans*, *Fusarium oxysporum*, *Penicillium funiculosum*, *sterile mycelium.*	*Aspergillus terreus*, *Aureobasidium pullulans*, *Choanephora cucurbitarun*, *Colletotrichum falcatum*, *Cylindrocarpon* sp., *Drechslera tetramera*, *Myrothecium roridum*, *Nigrospore sphaerica*, *Phoma hibernica*, *Rhizopus stolonifer*, *Trichoderma viride.*
	Winter Season	*Aspergillus flavus*, *Cladosporium cladosporioides*	*Aspergillus niger*, *Alternaria alternata*	*Aureobasidium pullulans*, *Nigrospora sphaerica*	*Arthrinium cuspidatum*, *Penicillium chysogenum*, *Phoma hibernica*, *Colletotrichum falcatum*, *sterile mycelium*	*Aspergillus luchuensis*, *Curvularia lunata*, *Aspergillus nidulans*, *Epicoccum nigrum*, *Fusarium oxysporum*, *Monilia geophila*, *Myrothecium roridum*, *Penicillium funiculosum*,
	Summer Season	*Airepnasodoi pullulans*	*Epicoccum nigrum*	*Alternaria alternate*, *Cladosporium cladosporioides*, *Aspergillus nidulans*, *sterile mycelium*	*Aspergillus flavus*, *A. niger*, *A. sulphureus*, *Curvularia lunata*, *Drechslera tetramera*, *unidentified Basidiomcete.*	*Aspergillus luchuensis*, *Choanephora cucurbitarum*, *Nigrospora sphaerica*, *Penicillium chrysogenum*, *P. funiculosum*, *Rhizopus stolonifer.*
Yellow Leaves which were recently fallen on the ground (YF)	Rainy Season	*Aspergillus luchuensis*, *Gloeosporium* sp.	*Aspergillus fumigatus*, *A. niger*, *Phaeoramularia graminicola*, *Penicillium charysogenum*	*Alternaria alternate*, *Fusarium equiseti*	*Aspergillus flavus*, *Colletotrichum falcatum*, *Curvularia lunata*, *Cylindrocarpon* sp., *Drechslera tetramera*, *Emericellan nidulans*, *Penicillium funiculosum*, *sterile mycelium.*	*Aspergillus sulphureus*, *Choanephora cucurbitarum*, *Aureobasidium pullulans*, *Cladosporium cladosporioides*, *Fusarium oxysporum*, *Myrothecium roridum*, *Penicillium nigricans*, *Phoma hibernica*, *Rhizopus stolonifer*, *Trichoderma viride.*
	Winter Season	*Cladosporium cladosprioides*	*Aspergillus niger*, *Aureobasidium pullulans Gloeosporium* sp.	*Aspergillus flavus*	*Colletotrichum falcatum*, *Penicillium chrysogenum*, *Rhizopus stolonifer*	*Aspergillus sulphureus*, *A. terreus*, *Arthrinium cuspidatum*, *Alternaria alternata*, *Curvularia pallescens*, *Epicoccum nigrum*, *Fusarium semitectum*, *Mucor hiemalis*, *Nigrospora sphaerica*, *Penicillium funiculosum*, *Phoma hibernica*, *Sterile mycelium*, *Trichoderma viride*
	Summer Season	*None*	*Fusarium equiseti*, *Fusarium semitectum*, *Gloeosporium* sp.	*Aspergillus niger*, *Fusarium lateritium*, *Myrothecium roridum*, *sterile mycelium*	*Aspergillus flavus*, *A. fumigatus*, *A. sulphureus*, *Alternaria alternate*, *Collectotrichum falcatum*, *Aspergillus nidulans*, *Phoma hibernica.*	*Aspergillus humicola*, *A. luchuensis*, *Aureobasidium pullulans*, *Cladosporium cladosporioides*, *Curvularia lunata*, *Fusarium oxysporum*, *Mucor hiemalis*, *Rhizoctonia solani*, *Rhizopus stolonifer*, *Trichoderma viride*
Partially Decomposed Leaves (PD)	Rainy Season	*Colletotrichum falcatum*	*Aspergillus flavus*, *A. niger*	*Aspergillus fumigatus*, *A. luchuensis*	*Fusarium oxysporum*, *Penicillium chrysogenum*	*Alternaria alternata*, *Choanephora cucurbitarum*, *Cylindrocarpon* sp., *Drechslera tetramera*, *Aspergillus nidulans*, *Gloeosporium* sp., *Mucor hiemalis*, *Penicilium funiculosum*, *Rhizopus stolonifer*, *Trichoderma viride.*
	Winter Season	*Cladosporium cladosporioides*	*Aspergillus flavus*, *Mucor hiemalis*	*Aspergillus niger*, *A. Sulphureus*, *Aureobasidium pullulans*, *Gloeosporium* sp.,	*Phoma hibennica*, *Tnichodenma vinide*	*Alternaria alternate*, *Collectotrichum falcatum*, *Curvularia lunata*, *Fusarium semitectum*, *Penicillium chrysogenum*, *Rhizopus stolonifer*, *sterile mycelium.*
	Summer Season	*Aspergillus fumigatus*, *A. luchuensis*, *A. niger Fusarium lateritium*, *Myrothecium roridum*, *Absidia repens*	*Alternaria alternate*	*Aspergillus flavus*, *Cladosporium cladosporioides*, *Aspergillus nidulans*	*Aspergillus sulphureus*, *Aureobasidium pullulans*, *Collectotrichum falcatum*, *Monilia geophila*, *Mucor hiemalis*, *Penicillium citrinum*	*Aspergillus terreus*, *Curvularia lunata*, *Fusarium semitectum*, *Nigrospora sphaerica Rhizopus stolonifer*, *sterile mycelium*, *Trichoderma viride*
Completely Decomposed Leaves (CD)	Rainy Season	*Aspergillus niger*, *Colletotrichum falcatum*, *Didymium nigripes*, *Myrothecium roridum*	*Aspergillus flavus*, *Fusarium oxysporum*	*Fusarium semitectum*, *Penicillium chrysogenum*	*Aspergillus fumigatus*, *A. luchuensis*, *Mucor hiemalis*, *Rhizopus stolonifer*, *Trichoderma viride*, *sterile mycelium.*	*Choanephora cucurbitarum*, *Aspergillus nidulans*, *Phaeoramularia graminicola.*
	Winter Season	*Aspergillus flavus*, *A. Sulphureus*, *Cladosporium cladosporioides*	*Aspergillus niger*, *Mucor hiemalis*	*Rhizopus stolonifer*	*Alternria alternate*, *Aspergillus luchuensis*, *Fusarium semitectum*, *Memnoniella echinata*, *sterile mycelium*, *Trichoderma viride.*	*Stachybotrys atra*
	Summer Season	*Aspergillus flavus*, *Cladosporium cladosporioides*, *Fusarium lateritium*, *Myrothecium roridum*, *Phoma hibernica*, *Colletotrichum falcatum*	*Aspergillus niger*, *A. sulphureus*, *sterile mycelium*	*Aspergillus fumigatus*, *A. luchuensis*, *Aspergillus nidulans*, *Fusarium oxysporum*, *F. semitectum*	*Absidia repens*, *Aspergillus terreus*, *Mucor hiemalis*, *Penicillium citrinum*, *Rhizopus stolonifer*	*Aspergillus parasiticus*, *Alternaria alternata*, *Bipolaris* sp., *Trichoderma viride.*

## Data Availability

Not applicable.

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
