# Peer review of "Diversity, Succession and Seasonal Variation of Phylloplane Mycoflora of Leucaena leucocephala in Relation to Its Leaf Litter Decomposition"

_jof, 2022, doi:10.3390/jof8060608_

Round 1

Reviewer 1 Report

Dear authors, the manuscript entitled "Diversity, Succession and Seasonal Variation of Phylloplane Mycoflora of Leucaena leucocephala in Relation to its Leaf Litter Decomposition" presents an interesting topic for the current context of a better understanding of the decomposition mechanism and biological function of fungal components.

There are some suggestions that I consider will improve your work.

General suggestion - avoid using "we", "our" or any personal style of writing. Make the sentences where it appears more impersonal.

pay attention to references in text - line 37 [1}. line 72 phylloplane [20-31]. I think here is 30-31. Please change As per to According to (line 89, line 93)

Abstract - lines 13-17 - please make a condensed form of these sentences, which present general information. Add some more of your findings.

Introduction - the overall assessment of the Introduction is good, but it requires some changes in the last paragraph. Present a clear aim of your work and the objectives/hypotheses in a separate sentence each. This will help the reader to navigate and understand your work.

Mat and Meth section. Add the coordinates of the sampling areas. Add the number of collected leaves

Add a Data analysis section and use your results from each type of leaf to make statistics of your results. ANOVA and a multiple comparison test will improve your work. You have six categories of leaves analyzed throughout an entire year. This gave you a lot of results to make good statistics.

Results section

The tables appear to be very large, try to make a condensed form. In this form, they looks like a report. If you want to keep in text, they should be explained in detail. You present the Average in each table? Do not repeat the same information from table in a graph. Chose only one. Figure 1, you speak about 100% maximum. Change the y axis to 100%, not to 120%. Add a note below the table to detail the number of samples analyzed.

Line 152-153 - This sentence is not necessary. You present every type of leaf separately. 

Line 165-186 need a reference to a table, or figure.

Figure 3,5,7,9,11 - idem figure 1 120 to 100% y axis

The results section need a deep change. You do not provide enough text for each of the presented tables. Try to make sub-sections for each of the leaves type and a sub-section for seasonal variation.

Your data are interesting and deserve to be well presented.

The discussion section is interesting and well written.

The conclusion presents well your main findings.

The article presents an interesting study. It should be reorganized, the Results part, a process that will give the opportunity to the authors, if they consider, to add more references in the discussion section and an improvement conclusion

Reviewer 2 Report

As general comment the work is well written and designed with relevant results.

In general terms the topic of the article is interesting, the methodology is explicitly presented and the results reported are interesting.

The structure of the paper is correct.

In my opinion, the abstract is too general, please reframe.

The introduction chapter should end with a paragraph indicating the purposefulness of the conducted research. Authors should clearly define the purpose of the work and formulate research hypotheses.

Materials and method section is well described and correspond to the aim set out in the manuscript. 

The tables and figures clearly presenting the obtained results with their appropriate interpretation.

The references are sufficient and necessary.

My comments are related to the current nomenclature of fungi (according to Index Fungorum).

Currently:

- No Emericella nidulans (Eidam) Vuill. but Aspergillus nidulans (Eidam) G. Winter.

- No Rhizopus nigricans Ehrenb. but Rhizopus stolonifer (Ehrenb.) Vuill.

etc. in all text.

The paper needs some editorial corrections.

I recommend the publication of this manuscript in the Journal of Fungi journal after minor revisions.

Reviewer 3 Report

This is an interesting manuscript about the fungal colonization of Leucaena leucocephala leaves at various stages of decomposition and their use as green manure. In total, fifty-two different species of fungi were isolated using various techniques with a higher percentage occurrence of Deuteromycetes (75.47%) and a lower rate of Ascomycetes (9.43%). This manuscript may be acceptable as a limitedly focused paper. I have one comment before accepting this manuscript.

- Please add some details about the identification of the fungi in this study whether morphological or molecular identification.

Round 2

Reviewer 1 Report

Dear authors, you have improved your work and this form of the manuscript presents better your findings.